# Evaluation of Soil and Ambient Air Pollution Around Un-reclaimed Mining Bodies in Nižná Slaná (Slovakia) Post-Mining Area

**DOI:** 10.3390/toxics8040096

**Published:** 2020-10-29

**Authors:** Lenka Demková, Július Árvay, Lenka Bobuľská, Martin Hauptvogl, Miloslav Michalko, Jana Michalková, Ivona Jančo

**Affiliations:** 1Department of Ecology, Faculty of Humanities and Natural Sciences, University of Prešov, 17. Novembra 1, 08001 Prešov, Slovakia; lenka.bobulska@unipo.sk; 2Department of Chemistry, Faculty of Biotechnology and Food Sciences, Slovak University of Agriculture in Nitra, Tr. A. Hlinku 2, 94976 Nitra, Slovakia; julius.arvay@gmail.com (J.Á.); ivonajanco@gmail.com (I.J.); 3Department of Sustainable Development, Faculty of European Studies and Regional Development, Slovak Agricultural University in Nitra, 94976 Nitra, Slovakia; martin.hauptvogl@uniag.sk; 4Department of Geography and Applied Geoinformatics, University of Prešov, 17. Novembra 1, 08116 Prešov, Slovakia; miloslav.michalko@unipo.sk (M.M.); jana.michalkova@unipo.sk (J.M.)

**Keywords:** former mining area, *L. pseudoscabrum*, moss and lichen bag technique, soil contamination, contamination factor, degree of contamination, relative accumulation factor

## Abstract

Thirty soil samples were taken, and the same number of moss (*Dicranum scoparium*) and lichen (*Pseudevernia furfuracea*) bags were exposed to detect environmental pollution in the former mining area Nižná Slaná. Soil and ambient air are influenced by hazardous substances, which leak from old mining bodies due to insufficient or completely missing reclamation. The total content of the risk elements (As, Cd, Co, Cu, Fe, Hg, Mn, Ni, Sb, Se, Pb, Zn) was determined in soil, moss, and lichen samples and in the bodies of *Leccinum pseudoscabrum*. Biological (soil enzymes—urease, acid phosphatase, alkaline phosphatase, fluorescein diacetate (FDA), ß-glucosidase) and chemical properties (pH) were determined in soil samples. Contamination factor (C_f_), degree of contamination (C_d_), pollution load index (PLI), and enrichment factor (EF) were used for soil and relative accumulation factor (RAF) for air quality evaluation. Contamination factor values show serious pollution by Cd, Fe, Hg, and Mn. Pollution load index confirmed extremely high pollution almost at all evaluated areas. Soil enzymes reacted to soil pollution mostly by decreasing their activity. Mosses and lichens show differences in the accumulation abilities of individual elements. Regular consumption of *L. pseudoscabrum* would provide the dose of Cd and Hg below the limit of provisional weekly intake. Based on the bioaccumulation index (BAF) values, *L. pseudoscabrum* can be characterized as an Hg accumulator.

## 1. Introduction

The environmental impacts of the spreading of hazardous elements from former mining bodies have been the object of numerous studies all over the world [1,2,3,4]. While some mining bodies represent only a potential risk (become dangerous when disrupted), others are a serious environmental problem causing the degradation of natural ecosystems. The danger of the tailing ponds lies in the dust consistency of sewage sludge, which is easily spreading by the wind at a long distance to the surrounding landscape. Small particles contain hazardous elements such as toxic metals, which are dangerous for the environment and human health. Numerous diseases caused by dust pollution were described in earlier studies [5]. A polluted environment causes defects in both animal and plant bodies [6,7,8]. Effective remediation can reduce the impact of hazardous elements on the environment, but due to the high cost, lack of interest from the government, or unsettled property issues a lot of tailing ponds are left to self-development [9]. A whole range of anthropogenic activities contribute to soil pollution by hazardous substances [10,11,12]. Mining activities and mining-related industries are among the leading ones. Ore mining is inevitably connected with the excavation of huge amounts of dump rocks that disrupt environment balance, cause irreversible changes in landscape structure, and are the source of undesirable substances releasing the soil, water, and air. The state of the soil quality is reflecting on soil conditions, which are responsible for soil fertility and its correct functionality [13].

Moss and lichen transplantation in the polluted sites as an airborne biomonitoring method is widely used [14]. This method has several advantages comparing in situ methods, such as low cost, opportunity to choose exposure time, and monitoring sites [15]. Moss and lichen taxa selection depends on the opportunities in different geographical conditions. Many previous studies deal with the differences between moss and lichen accumulation abilities [16,17]. The ability to accumulate different hazardous elements varies depending on moss/lichen taxa, environmental conditions and time of exposure [18,19].

Soil microbial community plays a key role in soil processes. Soil enzymes catalyze reactions in the soil system that have biochemical significance, participate in nutrient cycles, and transfer energy through organic matter decomposition; nutrients are released to be available for plant growth [20]. Soil enzymes are additionally used as a bioindicator of soil quality [21]; because they are very sensitive to environmental stress, they react quickly to the changes in the soil environment, and their determination is not difficult. The activities of urease (URE), acid phosphatase (ACP), and alkaline phosphatase (ALP), ß-glucosidase (BG), and fluorescein diacetate (FDA) were used in numerous studies to detect the influence of soil pollution to the soil quality [22,23].

The aims of the study are (i) to assess the soil pollution around the former mining bodies in Nižná Slaná mining area using indices (enrichment factor (EF), contamination factor (*C_f_*), degree of contamination (*C_d_*); pollution load index (PLI)), (ii) to determine the influence of soil pollution on the soil quality using soil enzyme activities, (iii) to estimate the ambient air pollution using moss and lichen biomonitoring method through relative accumulation factor (RAF), and (iv) to compare accumulation abilities of the two exposed taxa.

## 2. Materials and Methods

The former mining village Nižná Slaná is situated in the south-eastern part of Slovakia (Figure 1) along the Slaná river valley surrounded by the Slovenské rudohorie hills. Climatological features are typical for a moderately warm to moderately wet climate region, with an average January temperature of −2 °C to −5 °C. According to Köppen climate classification [24], the study area belongs to the warm-summer humid continental climate, with the coldest month average temperature below 0 °C (or −3 °C). All month’s average temperatures were below 22 °C, and at least four months averaging above 10 °C. There were no significant precipitation differences between seasons.

The village Nižná Slaná was established solely as the mining municipality in the 14th century, where the iron ores, precious metals, especially silver, pure mercury, and vermilion were mined. The biggest expansion of the mining activities occurred at the beginning of the 19th century what was related to the big iron smelter construction [25]. The mining activities continued until the beginning of the 20th century when all iron ores were exported for processing to the modern ironworks in Hungary. Nowadays, the whole area is included in the list of environmental burdens [9]. The next serious environmental burden in the village cadastre is a tailing pond localized in the northeast direction from the village. The tailing pond [48°44′36.9′’; 25°25′51.24′’] covers an area of 20.6 ha and around 7 million tons of the sludge is stored there. Sludge has the consistency of the fine-grained material that is, during the windy weather, transmitted to the surrounding villages. Due to the unclear ownership relations, the lack of reclamation and the rehabilitation of the dam, security, and stability of the tailing pond is questionable. In past studies, the danger of lung cancer among people under the regular pollution coming from Nižná Slaná mines was confirmed [26].

According to the International Union of the Soil Science recommendation, soils formed by the material of technogenic origins should be defined as the technosol [27]. In our study, sampled material consistency was changing depending on the distance and the direction from the main environmental loads (main mining bodies—tailing pond, iron processing plant), so we used the term “soil” for all the soil/technosol samples.

Soil samples were collected during summer 2018 in the Nižná Slaná village cadastre. Sampling sites were designed as a square network with a distance between two sites of 200 m (due to natural or other obstacles, this distance has not always been fully maintained). The sampling sites network cover the area of the tailing pond, surrounding of the ironwork, and the area which is considered as the most affected by the dust coming from the tailing pond (in the prevailing wind direction) (Figure 1).

In total, 30 soil samples of 500 g of the topsoil (0–10 cm), were sampled from each sampling site, stored in the plastic bags and transported to the laboratory. One part of each sample was frozen (−18°C) and later used for enzyme activity analysis. The second part was dried at room temperature, handy crushed, cleaned from the roots and dead plant parts, and finally sieved through a mesh sieve (2 mm).

The total content of the hazardous elements As, Cd, Co, Cu, Fe, Hg, Mn, Ni, Sb, Se, Pb, and Zn was determined in the soil also in moss and lichen samples. Elemental analysis was carried out on an Agilent ICP-OES spectrometer 720 (Agilent Technologies Inc., Santa Clara, CA, USA) with axial plasma configuration and with an auto-sampler SPS-3 (Agilent Technologies, GmbH, Waldbronn, Germany). Assessment of the hazardous elements in the soil, moss, and lichen samples was described several times in detail in earlier studies [17,23,28,29]. This procedure was maintained for this study.

Exchangeable soil pH was measured by pH meter inoLab pH 720 in a solution of 20 g of soil and 50 mL of CaCl_2_ (c = 0.01 mol L^−1^) (Sigma-Aldrich, Ltd., Bratislava, Slovakia). The activity of urease was determined by a spectrometer as an ammonium release (at 410 nm) after 24 h incubation (at 37 °C) of soil samples with a urea solution [30]. The activity of acid and alkaline phosphatase was measured by a spectrometer as a phenol release (at 510 nm) after 3h of incubation (at 37 °C) of soil samples with a phenyl phosphate solution and acetate buffer (pH = 5) for acid phosphatase and borate buffer (pH = 10) for alkaline phosphatase [31]. Fluorescein diacetate activity (FDA) was measured by the spectrometer at 490 nm after the 1h of incubation at 30 °C using fluorescein diacetate as a soil after soil hydrolysis [32]. The activity of ß-glucosidase (BG) was spectrophotometrically determined (464 nm) as a p-nitrophenol release after the 1h incubation at 37 °C of the soil samples with 4-nitrophenyl glucopyranoside [33].

Contamination factor (*C_f_*) described by [34] was used to express the level of soil pollution by individual hazardous elements. The calculation was conducted as follows Equation (1):(1)$$C_{f}^{i}=\frac{C_{0-1}^{i}}{C_{n}^{i}}$$
where: $C_{0-1}^{i}$ is the total concentration of the hazardous element in soil and $C_{n}^{i}$ is the background level of the hazardous element. Background levels for As, Cd, Co, Cu, Fe, Hg, Mn, Ni, Pb, Sb, Se, and Zn obtained from [35,36] were are 7.20, 0.30, 60.0, 19.0, 530, 0.08, 20.0, 25.0, 25.0, 1.04, 0.10, and 65.0 mg kg^-1^, respectively. Hakanson (1981) [34] divided contamination factor values into 4 classes: low contamination factor (if $C_{f}^{i}$ < 1), (ii) moderate contamination factor (if 1$\;\leq C_{f}^{i}\;$< 3); (iii) considerable contamination factor (if 3$\;\leq C_{f}^{i}\;$< 6) and (iv) very high contamination factor (if ≥
$C_{f}^{i}$6). Overall contamination of the study area by individual elements was expressed by the degree of contamination (*C_d_*) which was used to express the overall contamination of the study area by individual elements, and was calculated as follows Equation (2):(2)$$\mathrm{Cd}=\sum^{\text{​}}C_{f}^{i}$$

According to Hakanson [29], four classes of the degree of contamination *C_d_* are intended: (i) low degree of contamination (if *C_d_* < 8), (ii) moderate degree of contamination (if 8 ≤ *C_d_* < 16), (iii) considerable degree of contamination (if 16 ≤ *C_d_* < 32), and (iv) very high degree of contamination (if *C_d_* ≥ 32). The comprehensive level of soil pollution at each sampling site was expressed by the pollution load index (PLI) calculated according to [32] as follows Equation (3):*PLI* = *(C_f1_* × *C_f2_* × *C_f3_* × …. × *C_fn_)1/n*(3)
where: n is a number of assessed elements, ***C_f_*** is a contamination factor of individual pollutants. Wang [37] divided the *PLI* values into four classes as follows: (i) no pollution (*PLI* < 1), (ii) moderate pollution (1 ≤ *PLI* < 2), (iii) heavy pollution (2 ≤ *PLI* < 3), (iv) extreme pollution (*PLI* ≥ 3).

Enrichment factor (*EF*) is an indicator used to assess the presence and intensity of anthropogenic contamination on the surface soil. Enrichment factor for each of the risk elements in the substrate was calculated as follows Equation (4):(4)$$\mathrm{EF}\;=\;\frac{\left(\frac{C_{x}}{C_{ref}}\right)sample}{\left(\frac{C_{x}}{C_{ref}}\right)background}$$
where: *C_x_**/**C_ref_* is the ratio of the concentration of the hazardous element in the substrate sample to the concentration of the reference element. In this study, the reference metal was selected based on the correlation coefficient analysis from risk elements that are neither likely to be affected by anthropogenic activities not correlated with heavy metal pollutants. The background values of the elements were obtained from studies of [35,36]. Background values of As, Cd, Co, Cr, Cu, Fe, Hg, Mn, Ni, Pb, Sb, Se, Sr, Zn are 7.2, 0.30, 0.30, 60, 19, 520, 0.08, 20, 25, 25, 1.04, 0.10, 130, and 65, respectively. Enrichment factor values are divided into 5 classes (i) *EF* < 2 deficiency to minimal enrichment; (ii) 2 ≤ *EF* < 5 moderate enrichment; (iii) 5 ≤ *EF* < 20 significant enrichment; (iv) 20 ≤ *EF* < 40 very high enrichment; (v) *EF* ≥ 40 extremely high enrichment.

One moss (*Dicranum scoparium*) and one lichen (*Pseudevernia furfuracea*) taxa were collected during May 2018 in Čergov Mts (East Slovakia). Increased conservation attention of Čergov Mts. because of specific plant and animal species occurrence is a prerequisite for low environmental contamination. Additionally, no significant source of pollution (industrial factories, etc.) is localized in the surrounding area. The sites of moss and lichen sampling were localized at least 1000 m from the main and at least 500 m from the forest roads. Approximately 500 g of each taxon was collected, stored in a paper bag, and transported to the laboratory where the samples were manually cleaned from the dead parts, needles, and soil particles. Moss and lichen were washed three times (immersed to the water) in deionized water, lasting 5, 10, and 20 min (approximately 10 L water per 100 g of moss and lichen dry weight). After the washing, samples were hand-squeezed and air-dried in an over at 40 °C for 24 h (Venticell 111, BMT, a.s., Czech Republic).

Nylon net (2 mm) was cut at pieces (10 × 10 cm) where approximately 5 g of each taxon was wrapped and tied. One moss and one lichen bag were stored in the laboratory as a control sample (for the initial concentration determination). Each taxon, in two replicates was exposed in 30 sampling sites (in total 120 samples—two taxa, in two replicates in 30 sampling sites). Moss and lichen exposition sites were identical to the soil sampling sites. Bags were exposed on the trees or scrubs, approximately 2 m high, hanging outside the tree (at the end of the branch). After 8 weeks of the exposition, the samples were collected and analyzed for the risk elements concentration.

For elemental analysis, an Agilent ICP-OES spectrometer 720 (Agilent Technologies Inc., Santa Clara, CA, USA), with axial plasma configuration and with an autosampler SPS-3 (Agilent Technologies, GmbH, Waldbronn, Germany), was used. As mentioned above, the methodology of hazardous element determination was described in detail in earlier studies [17,23,28,29].

Relative accumulation factor (*RAF**)* was used to assess the content of each hazardous element in exposed mosses and lichens. *RAF* was calculated as follows Equation (5):*RAF* = (Cexposed − Cinitial)/Cinitial(5)
where *C_exposed_* is the content of the hazardous element after exposure and *C_initial_* represent the content of the hazardous element before exposure (not exposed sample—control).

The content of toxic elements was studied also in the mushroom species *Leccinum pseudoscabrum* (Kallenb.) Šutara (*n* = 27). Mushrooms were sampled during summer 2018, in parallel with the soil sampling and moss exposition. The occurrence of mushrooms was accidental in the evaluated area. The health risk from the mushroom consumption was evaluated based on the exposure to the Cd and Hg that was compared with the provisional tolerable weekly intake (PTWI). The PTWI set for Cd and Hg are 0.0062 and 0.004 mg kg^−1^ body weight (BW), respectively [38,39]. The exposure to the Cd and Hg was calculated as follows Equation (6):(6)$$E=\frac{C_{i}\;\times \;\mathrm{intake}}{BW}$$
where *E* is exposure, $C_{i}$ is a concentration of the element i, intake stands for the consumption of the mushrooms per week (based on the [40] that is 0.23 kg of fresh matter per week), and BW is body weight (70 kg for an average adult). The amount of fresh weight was calculated based on the assumption that dry matter represents 10% in mushrooms [41].

Bioaccumulation factor (BAF) indicates the ability of mushrooms to uptake elements from soil/substrate to their above-ground parts. BAF is calculated as follows Equation (7) (mg kg^-1^ of dry matter):(7)$$BAF=\frac{C_{i}\;mushroom}{C_{i}\;substrate\;}$$
where $C_{i}$ is the measured content of the element i. Based on Barker [42] the mushrooms can be divided into three groups: accumulators (*BAF* > 1), indicators (*BAF* = 1), and excluders (*BAF* < 1).

The map outputs were processed using the open-source Geographic Information System (GIS) using software QGIS (version 2.18) and open data from OpenStreetMap contributors [43]. All statistical analyses were performed in R studio [44]. Data were log-transformed prior to analysis. The non-parametrical Mann–Whitney U test was used to compare significant differences in accumulation abilities between taxa.

## 3. Results

### 3.1. Hazardous Element Content in Soil Samples

The values of hazardous elements in soils determined at 30 sampling sites expressed by descriptive statistics are listed in Table 1. Data were subsequently used for the indices calculation to express the state of pollution, the spreading of pollution, and the most contaminated sites in the study area. From the Table 1 it is clear, that As, Cd, Fe, Hg, Mn, and Sb exceeded the limit values at all 30 sampling sites, while Co, Cu, Ni, Pb, and Se exceeded the limit values only on some of them. The limit value for Cr and Zn was not exceeded in the study area. The area was a long time known as Fe, Hg, and precious metals deposit [45]. Dangerous pollution by As was confirmed in the study area by earlier studies [46].

### 3.2. Contamination Factor, Degree of Contamination and Pollution Load Index

The contamination factor (*C_f_*) which was calculated for each sampling site and each hazardous element confirmed that sampling sites are low in their contamination by Cr, low to moderately contaminated by Zn, considerably to very highly contaminated by Co, moderately to very highly contaminated by Sb, low to very highly contaminated by As, Cu, Ni, Pb, Se, and very highly contaminated by Cd, Fe, Mn, and Hg (Table 2). The level of contamination by individual elements in the study area expressed by the contamination factor was displayed by heatmaps (Figure 2).

As mentioned above, the area was the most seriously contaminated by Fe, Mn, and Hg with the centers of pollution localized in the body of the tailing pond and in the area of the former ore processing plant (sampling site no. 19 and 20). Serious levels of Cd were also confirmed around the processing plant. Regarding the heatmap, serious soil pollution appeared on sampling site no. 27, localized at the same altitude as the tailing pond but on the other side of the valley. According to Zobeck and Van Pelt [48], windborne transport and atmospheric deposition may play a significant role in spreading contaminants. Prevailing winds in the smelting operation areas control the transport and could take away the hazardous elements over a long distance [49].

Degree of contamination (Cd) is used for the overall assessment of the contamination in the study area. According to the degree of contamination values (Cd), the study area is very highly contaminated by As, Cd, Co, Cu, Fe, Hg, Mn, Ni, Pb, Sb, and Se; considerably contaminated by Zn; and moderately contaminated by chromium.

The pollution load index (PLI) was used to express the level of pollution at each sampling site considering individual elements’ contamination factor values. The state of the pollution in the study area expressed by PLI is visualized by heatmap (Figure 3). Almost all study area was, based on the PLI index values, evaluated as extremely polluted. The most serious pollution was determined (the same as in the case of contamination factor values) in the area of the tailing pond and around the sampling site no. 27.

### 3.3. Enrichment Factor Values in the Soil Samples

Values of *EF* lower than 2 (0.3 on a log scale) suggest that the trace element concentration comes from natural sources [50], but higher values of *EF* suggest an anthropogenic origin of the elements [51]. The most serious situation was determined for Fe (min–max: 33.9–934) and Mn (28.6–2140), where enrichment factor reached very high to extreme values at all sampling sites (Figure 4).

Because the area is well known as the former iron mining area, high values of iron were expected. In the case of As (3.79–899), Co (19.5–516), and Sb (2.59–289), moderate to extreme enrichment was determined. The enrichment factor of Cd (13.5–661) and Hg (8.47–908) reached significant to extreme values. The values of Cr (0.24–2.40), Se (0.08–202), Zn (0.47–13.9), and Ni (0.39–19.4) ranged between minimal and moderate or minimal and significant values. Enrichment factors for Cu (1.42–33.8) and Pb (0.56–24.5) ranged between minimal and very high values.

The activity of soil enzymes varied depending on the distance from the tailing pond and the prevailing wind orientation which brings toxic substances from the tailing pond. According to the meteorological measurements [52], the wind is heading from northwest to southeast, which is reflected in more significant pollution in the south-eastern part of the monitored area (Figure 5). Soil enzyme activities are greatly influenced by soil properties and could be significant indicators of heavy metal toxicity [22]. It has been repeatedly shown that risk elements’ toxicity could negatively affect soil enzymes which is manifested mainly by their reduced activity [53]. In previous studies, soil enzymes have been found to react differently to individual metal contamination. According to Acosta Martinez and Tabatabai [54], lead and cadmium have the greatest inhibitory effect on the activity of soil enzymes. Urease activity is significantly reduced by high lead content [55]. Copper has a significant inhibitory effect on β-glucosidase activity and arsenic on both urease and phosphatase activity [56]. Numerous studies have focused on others soil parameters that influence the activity of soil enzymes [20,57]. In addition to the content of risk elements in the soil, the activity of soil enzymes is also affected by temperature, humidity, soil pH, organic matter, nutrient availability, and chemical properties of the litter [58].

The lowest values of all evaluated soil enzymes were determined at sampling site 2, which is the body of the tailing pond. With the increasing distance from the tailing pond activity of the soil enzymes increase, with the exception of the most polluted area (regularly polluted by sewage sludge of dust consistency transmitted by the wind). The values of soil reaction varied between extremely acid and alkaline [35]. It has been documented in several studies all over the world that sewage sludge pH values used to reach strongly alkaline [59] to neutral values [60]. In accordance with the studies, strongly alkaline soil reaction was determined in the body of the tailing pond, and slightly acid to neutral values were assessed in the most polluted part of the study area.

### 3.4. Hazardous Element in the Ambient Air of Nižná Slaná

The contents of the hazardous elements in moss and lichen expressed by RAF values are listed in Figure 6. Based on the results we can conclude that lichen is a better accumulator of all evaluated hazardous elements, except Hg. Cecconi et al. [61] have confirmed excellent accumulation abilities of *Pseudevernia furfuracea*, which is why it is often used in bioaccumulation studies. Adamo [62] confirmed that lichens often accumulate higher concentrations of the hazardous element than mosses. However, because of easier accessibility, mosses are still more popular in bioaccumulation studies [61]. As mentioned above, only Hg values reached higher contents in moss compared to lichen. In the study of Shao et al. [63], who analyzed the content of Hg in mosses and lichens, higher values were found in lichen tissues. Additionally, the correlation between the altitude and the Hg content in the moss samples was also confirmed in this study.

Comparing average *RAF* values (regardless of the taxa) of individual elements we can conclude that pollution by hazardous elements decreased in the following order: Co>Se>Sb>As>Cd>Cr>Ni>Fe>Pb>Hg>Cu>Zn (Figure 7).

A non-parametrical Mann–Whitney U test was used to verify significant differences between taxa in their accumulation abilities (Table 3).

Significantly higher ability to accumulate Hg was found for moss (*Dicranum scoparium*). Lichen (*Pseudevernia furfuracea*) was found as a significantly better accumulator of Cd, Co, Cr, Fe, Mn, Pb, and Zn. For As, Cu, Ni, Sb, and Se, no significant differences between evaluated taxa were confirmed. Bargagli et al. [64] have found a higher concentration of lithophile elements (Al, Cr, Fe, Mn, Ni, and Ti) in moss and atmophile elements (Hg, Cd, Pb, Cu, V, and Zn) in lichens. On the other hand, Loppi and Bonini [65] confirmed higher values of Hg and Zn in moss samples comparing mosses. Because the mosses and lichens exhibit different accumulation abilities of individual elements, in many studies mosses and lichens are used together to reach comprehensive results [66].

### 3.5. BAF Evaluation

The genus *Leccinum* is characterized by the high variability of BAF values (Table 4). Jarzyńska and Falandysz [67] and Medyk et al. [68] reported that the genus *Leccinum* have BAF > 1 for Ag, Cd, Cu, Hg, K, Mg, Na, P, Rb, and Zn and BAF < 1 for Al, Ba, Ca, Cr, Fe, Mn, Ni, Pb, and Sr. The BAF values indicate that *L. pseudoscabrum* accumulates only Hg (Table 2). In general, the genus *Leccinum* is reported to have BAF values of Hg more than 1 [69].

### 3.6. Health Risk Assessment

The genus *Leccinum* is a popular and valuable wild-grown edible mushroom used traditionally as a gourmet food in Central and Eastern Europe and Scandinavia [60,67,70,71]. If we assume the consumption of 23 g fresh matter of *L. pseudoscabrum* per week, the intake of Cd and Hg would represent 5.9% and 47.4%, respectively, of the PTWI set for these toxic elements. The loss of Hg when the mushrooms are cooked is relatively low (around 10% when blanching for 10 min.) [72]. There is an increased risk of potentially harmful exposure to Hg, due to the fact, that almost 50% of the PTWI for Hg is reached by the consumption of a very small amount (23 g per week) of the mushrooms. Other foods that are consumed in much higher quantities would considerably increase the health risk. The average consumption of mushrooms would not represent a high health risk related to the Cd intake and thus can be considered safe.

## 4. Conclusions

Spreading of the hazardous elements from non-reclaimed mining bodies at the former mining area Nižná Slaná (Slovakia) was evaluated in soils and airborne air using biomonitoring methods. The hazardous elements’ values were expressed by factors and indices. The results of the contamination factor pointed out very high contamination of soils by Cd, Fe, Mn, and Hg and the pollution load index confirmed that almost the whole evaluated area is extremely polluted. Biological (soil enzymes) and chemical (soil reaction) soil properties react significantly to the environmental stress. The tailing pond and area of ore processing plants were found as the main sources of undesirable substances and the spreading of the pollution predominantly by wind caused serious pollution in the whole study area. The moss and lichen bag technique was used as the airborne biomonitoring method. While moss (*Dicranum scoparium*) showed significantly better ability to accumulate Hg, lichen (*Pseudevernia furfuracea*) was found as significantly suitable for Cd, Co, Cr, Fe, Mn, Pb, and Zn accumulation. Results of the BAF values indicate that *L. pseudoscabrum* is an accumulator of Hg. The average consumption of mushrooms can be considered as safe. However, whereas the study area also includes residential areas and pastures under the influence of dust emissions and agricultural lands with polluted soil, where agricultural crops are grown, we still consider the study area very dangerous in terms of health risk.

## Figures and Tables

**Figure 1 toxics-08-00096-f001:**
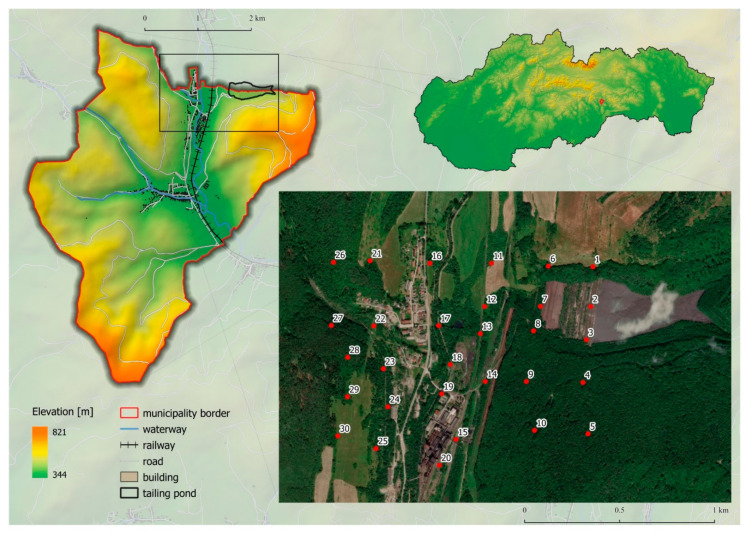
Study area and sampling points.

**Figure 2 toxics-08-00096-f002:**
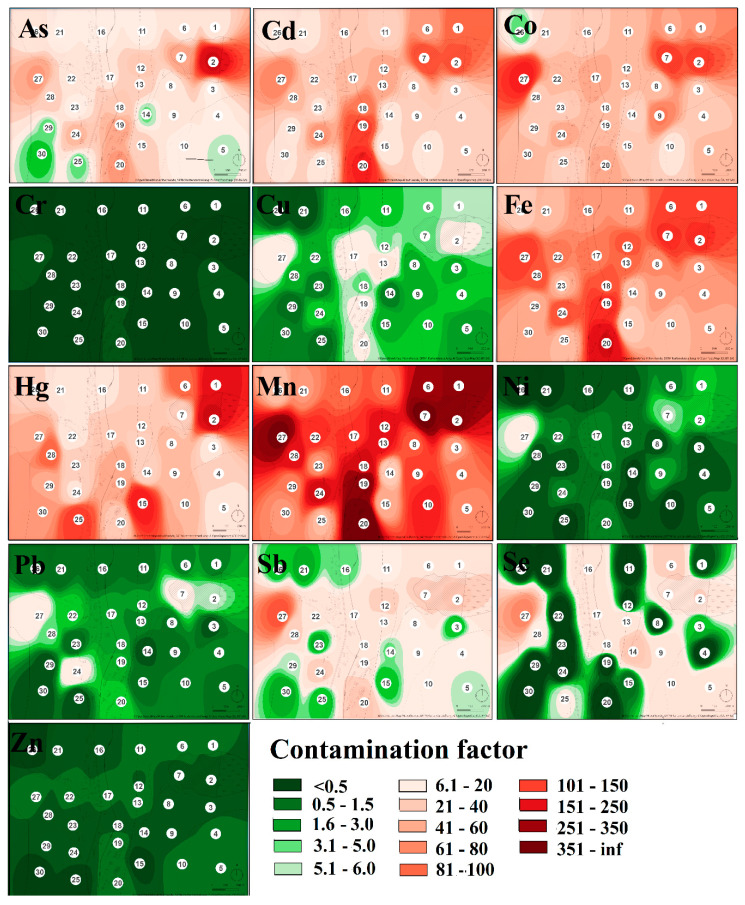
The heatmap of the contamination state in the study area expressed by contamination factor values.

**Figure 3 toxics-08-00096-f003:**
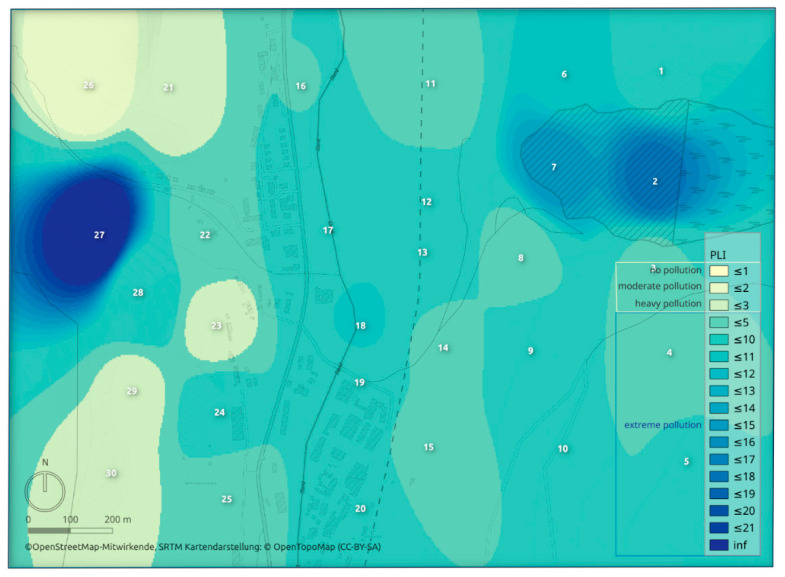
The pollution state in the study area expressed by the pollution load index (PLI).

**Figure 4 toxics-08-00096-f004:**
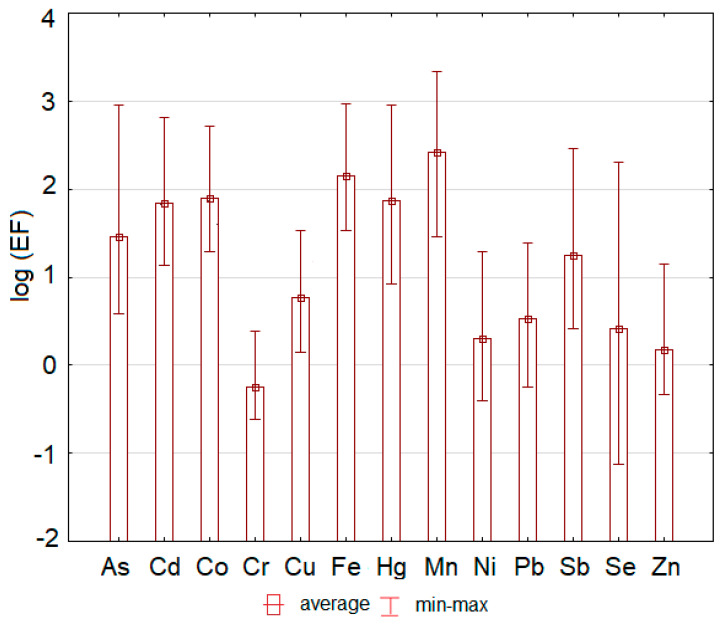
The hazardous elements enrichment factor values (expressed as the logarithm of the enrichment factor for better visualization) determined at each sampling site of the study area.

**Figure 5 toxics-08-00096-f005:**
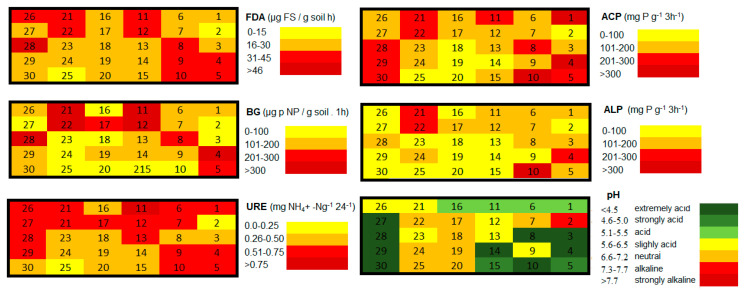
The values of soil enzymes (URE—urease; ALP—alkaline phosphatase, ACP—acid phosphatase, BG—ß-glucosidase, FDA—fluorescein diacetate) and soil pH determined at each sampling site of a study area.

**Figure 6 toxics-08-00096-f006:**
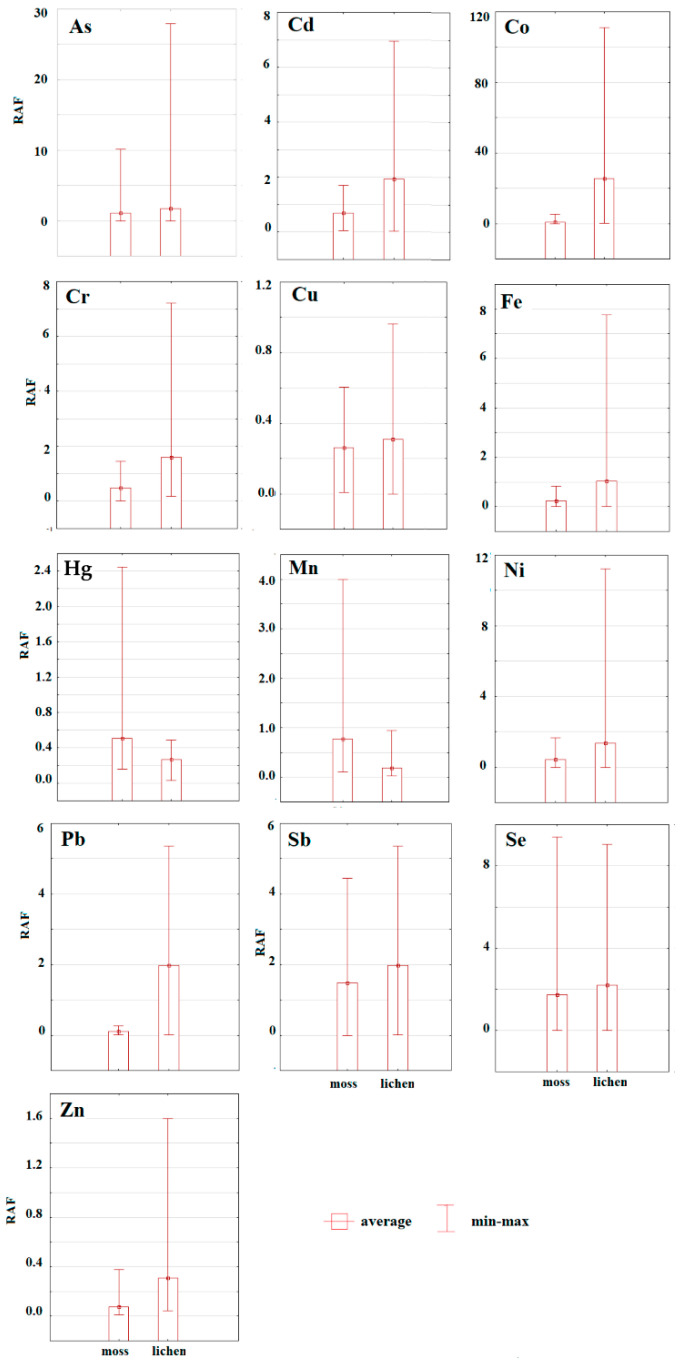
Differences between hazardous elements determined in moss and lichen taxa expressed by relative accumulation factor (RAF) values.

**Figure 7 toxics-08-00096-f007:**
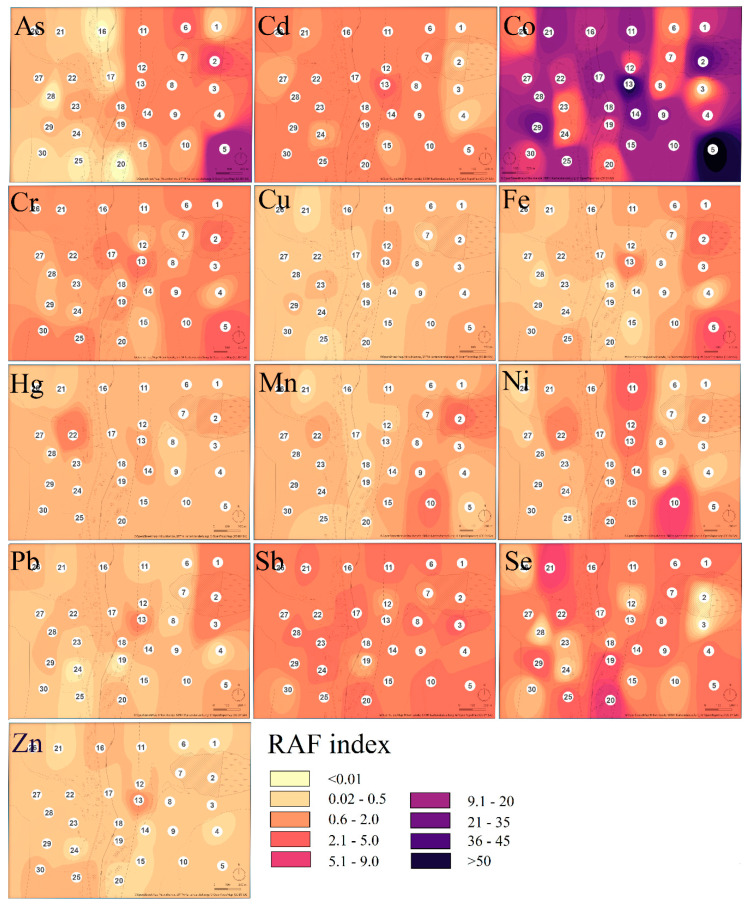
RAF values regardless of the taxa (average value) in the study area.

**Table 1 toxics-08-00096-t001:** Descriptive statistics for hazardous element values in soils (LV—limit values for Slovak soils, set by Act. No 220/2004 Coll of Laws [47] and background values (BV) according to Čurlik and Šefčík [35] and Kabata-Pendias [36].

Elements	As	Cd	Co	Cr	Cu	Fe	Hg	Mn	Ni	Pb	Sb	Se	Zn
	[mg kg^−1^ DW (dry weight)]												
min	26.2	3.17	1.35	4.58	14.2	16,330	0.71	675	3.33	12.9	2.62	0.01	16.3
max	1834	53.3	56.5	52.6	301	133,196	20.6	11,510	177	223	109	7.38	95.9
avg	203	14.8	15.7	21.2	76.4	47,352	4.73	3797	37.8	62.5	15.9	1.00	57.2
med	85.5	9.23	13.2	22.3	56.9	35,999	2.73	2986	24.8	42.9	10.1	0.21	55.3
stdev	329	12.1	11.4	10.5	57.8	26,913	4.91	2836	36.1	57.0	19.8	1.52	20.7
LV	25.0	0.70	15.0	70.0	60.0	550	0.50	550	50.0	70.0	0.70	0.40	150
BV	7.20	0.30	0.30	60.0	19.0	530	0.08	20.0	25.0	25.0	1.04	0.10	65.0

**Table 2 toxics-08-00096-t002:** Descriptive statistics expressing the values of the hazardous elements contamination factor (*C_f_*) and the degree of contamination (*C_d_*).

*C_f_*	As	Cd	Co	Cr	Cu	Fe	Hg	Mn	Ni	Pb	Sb	Se	Zn
min	3.64	10.6	4.49	0.08	0.75	30.8	8.87	33.8	0.13	0.52	2.52	0.01	0.25
max	25.0	178	188	0.88	15.9	251	257	575	7.09	8.93	105	73.8	1.48
avg	28.3	496	52.4	0.35	4.02	89.3	59.1	189	1.51	2.49	15.3	9.95	0.88
med	11.9	30.8	43.9	0.37	2.99	67.9	34.1	149	0.99	1.72	9.71	2.12	0.85
stdev	45.8	40.5	38.1	0.18	3.04	50.8	61.4	141	1.44	2.28	19.0	15.2	0.32
*C_d_*	847	1488	1571	10.6	120	2680	26.4	1774	5696	45.3	74.9	457	8.49

**Table 3 toxics-08-00096-t003:** Results of the Mann–Whitney U test to compare significant differences in accumulation abilities between taxa.

Element		U	Z	*p*-Value
As	Between taxa	408	−0.61	0.53
Cd		148	−4.46	0.001 **
Co		276	−2.55	0.011 *
Cr		158	−4.33	0.001 **
Cu		449	−0.01	0.99
Fe		300	−2.21	0.03 *
Hg		204	−3.62	0.001 **
Ni		389	−0.89	0.37
Mn		82	−5.43	0.001 **
Pb		138	−4.60	0.001 **
Sb		337	−1.66	0.09
Se		403	−0.68	0.49
Zn		224	−3.34	0.001 **

* *p* < 0.05; ** *p* < 0.01.

**Table 4 toxics-08-00096-t004:** Content of the studied elements (mg kg^−1^ DW) and their BAF values.

Element Content	As	Cd	Co	Cr	Cu	Fe	Hg	Mn	Ni	Pb	Sb	Se	Zn
*C_mushroom_*	2.77	1.66	0.02	0.12	19.2	46.6	5.77	17.4	1.32	0.25	0.84	2.66	73.3
*C_topsoil_*	107	4.88	11.7	22.6	44.9	49,517	4.35	3811	27.4	48.8	5.73	2.91	79.1
*BAF*	0.03	0.34	0.00	0.01	0.43	0.00	1.33	0.00	0.05	0.01	0.15	0.92	0.93

*C_mushroom_*: average content of the element in the mushroom, *C_topsoil_:* average content of the element in the topsoil.

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
