# Peer review of "Evaluation of Soil and Ambient Air Pollution Around Un-reclaimed Mining Bodies in Nižná Slaná (Slovakia) Post-Mining Area"

_toxics, 2020, doi:10.3390/toxics8040096_

Round 1

Reviewer 1 Report

Overall comment -

The manuscript aims to address the pollution levels in post-mining area in Slovakia. The title of the manuscript should be adjusted to reflect better the site-specific nature of the study. The quality and clarity of the manuscript is affected by presentation that the authors are recommended to check the English and the presentation thoroughly. As for the Figures, like 2 and 7, the sampling sites should be more visible to readers. The conclusion of "the are very dangerous also in terms of health risk" (line 245) is not consistent with the discussion of 3.6 Health risk assessment (line 314-324). 

Specific comments - 

Line 34-35 - referring to "The spreading of hazardous elements from former mining bodies was the objective of numerous studies all over the world", the spreading was probably not the objective of studies? 

Line 42-43 - to add supporting evidence 

Line 60-65 - there are more than one aim. It should read "The aims of the study are ....., and iv)...

Line 88 - pls add the rationale as to the different sampling time of soils in summer 2018 and moss/lichen in May 2018 

Line 100 - Figure 1 - pls put it into the context, e.g. scale?, add appropriate captions to show site/ location

Line 180-181 - pls clarify sampling of mushrooms was conducted in this study. If yes, pls elaborate the timing/location?

Line 317 - pls justify the assumption of consumption loading made 

Author Response

Dear reviewer, 

we would like to thank you, for all your comments and recommendations, which help to improve the manuscript quality.

Specific comments to your recommendations are listed below: 

  • The title of the manuscript should be adjusted to reflect better the site-specific nature of the study.
    The localization of the study was added to the title of the manuscript.
  • The quality and clarity of the manuscript is affected by the presentation that the authors are recommended to check the English and the presentation thoroughly.
    The English language was checked by the native speaker. The manuscript was properly checked by the authors to review the presentation.
  • As for the Figures, like 2 and 7, the sampling sites should be more visible to readers.
    Thank you for your comment. The numbers were rewritten to be more visible.
  • The conclusion of "the are very dangerous also in terms of health risk" (line 245) is not consistent with the discussion of 3.6 Health risk assessment (line 314-324). 
    Thank you! We rewrote the sentence to be more clear.

Specific comments - 

  • Line 34-35 - referring to "The spreading of hazardous elements from former mining bodies was the objective of numerous studies all over the world", the spreading was probably not the objective of studies? 
    Thank you. The sentence was rewritten.
  • Line 42-43 - to add supporting evidence 
    The evidence was added.
  • Line 60-65 - there are more than one aim. It should read "The aims of the study are ....., and iv)...
    Thank you. The sentence was rewritten.
  • Line 88 - pls add the rationale as to the different sampling time of soils in summer 2018 and moss/lichen in May 2018 
    Moss samples were sampled earlier because we needed to prepare "moss bags" which were used as bioindicators of air pollution. Moss bags were exposed in the soil sampling sites.
  • Line 100 - Figure 1 - pls put it into the context, e.g. scale?, add appropriate captions to show site/ location
    Thank you for your recommendation. The scales were added to the maps.
  • Line 180-181 - pls clarify sampling of mushrooms was conducted in this study. If yes, pls elaborate the timing/location?
    Information about sampling, timing and location was added to the manuscript.
  • Line 317 - pls justify the assumption of consumption loading made
    We just want to say that even though the ptwi was not exceeded when consumed, so a small amount of consumed mushrooms (only 23 g) got into the body through the consumption of mushrooms half of the tolerable amount of Hg. If the consumption of other foods is theoretically added to this, it is highly likely that tolerable intake will be exceeded.

Best regards, 

Reviewer 2 Report

In the article results of a survey carried out on the area of former mining area are presented. For estimation of air composition an active biomonitoring using moss bags was applied. In soil samples activity of selected enzymes and parameters describing their physico-chemical parameters were determined.
In moss, lichen, an edible mushroom and soil samples concentrations of the selected hazardous chemical elements were determined.
To describe pollution of the environment appropriate indexes and factors were calculated. High pollution level for Cd, Fe, Hg and Mn was concluded. A difference in lichen and moss cumulation ability was observed. No health threat related to the mushroom consumption was stated.

Remarks
1. In maps presented in Fig. 2 "bull's eye" effect (isoline surrounding a sampling point) is observed. What method was used for maps construction? Consider change of parameters used for maps drawing.
2. In Fig. 4 bars' colour masks 'min' whiskers. The bar fill should be changed or removed.
3. In header of Tab. 1 the material type (soil) should be mentioned. A reader shouldn't be compelled to look for explanation somewhere in text.
4. l.212, 311 - explain DW
5. l.311 upper index in mg kg-1
6. Gray bars in Fig. 6 are undesired. They suggest confusing information indicating negative RAF values.
7. Beside a stament that Supplementary materials are (will be?) under a mdpi URL, they are not mentioned in the article.
8. Careful verification of references is required. In line 405 Ref. 17 Acces to https://www.enviroportal.sk/environmentalne-zataze/ reports response status "Not Found" (404 - Stránka nenájdená), access 28.09.2020. Author's access date is required.
Some referenced do not fit calls in the text. For example, check R language [37], OpenStreetMap contributors [39], R studio [40].

Author Response

Dear reviewer, 

We would like to thank you for all your recommendations, which help to improve the quality of the manuscript. All corrections are marked in the body of manuscript by red color. I would like to point out, that because there were a lot of references added, all Reference part was rewritten and renumbered.

Specific comments to your recommendations are listed below:

Reviewer #2

In the article results of a survey carried out on the area of former mining area are presented. For estimation of air composition an active biomonitoring using moss bags was applied. In soil samples activity of selected enzymes and parameters describing their physico-chemical parameters were determined. 
In moss, lichen, an edible mushroom and soil samples concentrations of the selected hazardous chemical elements were determined.
To describe pollution of the environment appropriate indexes and factors were calculated. High pollution level for Cd, Fe, Hg and Mn was concluded. A difference in lichen and moss cumulation ability was observed. No health threat related to the mushroom consumption was stated.

Remarks
1. In maps presented in Fig. 2 "bull's eye" effect (isoline surrounding a sampling point) is observed. What method was used for maps construction? Consider change of parameters used for maps drawing.
We used Inverse Distance Weighted (IDW) interpolation method utilizes the expectation of positive autocorrelation in relatively evenly spaced point field. The analysis provided a raster data model with interpolated values for the entire studied area. Mainly due to the big range of sample data value in a relatively small area, the map doesn´t represent data in eye-candy way. Thus for the representation of the data, we decided for "discret zones" not "linear" visualisation method, because we want to show boundaries of potential spatial contaminations.

2. In Fig. 4 bars' colour masks 'min' whiskers. The bar fill should be changed or removed.
Thank you. Figure was changed according to your suggestion.
3. In header of Tab. 1 the material type (soil) should be mentioned. A reader shouldn't be compelled to look for explanation somewhere in text. 
Thank you. Table caption was supplemented.
4. l.212, 311 - explain DW
Thank you. The explanation was added to the Table 1.
5. l.311 upper index in mg kg-1
Thank you. The upper index was corrected.
6. Gray bars in Fig. 6 are undesired. They suggest confusing information indicating negative RAF values.
Thank you. Figure was changed according to your suggestion
7. Beside a stament that Supplementary materials are (will be?) under a mdpi URL, they are not mentioned in the article.
Thank you for your comment. It is a mistake. There are no supplementary materials in the article. The statement was deleted.
8. Careful verification of references is required. In line 405 Ref. 17 Acces to https://www.enviroportal.sk/environmentalne-zataze/ reports response status "Not Found" (404 - Stránka nenájdená), access 28.09.2020. Author's access date is required. 
Thank you. Reference was rewritten and the date was added.

Reviewer 3 Report

I like the article. The authors did a good research-work and properly presented the results of their studies. The use of both soil tests and airborne observations were appropriate. I have only a few comments to the article:
[line 35]: In my opinion, the sentence is too general. Please explain which ones are potential and which are serious.
[line 35]: [1-3] ... I propose to add examples of works from other post-mining environmental hazards, e.g. doi.org/10.1016/j.ijrmms.2020.104377.
[line 51]: I propose to add a reference to the described examples of mining hazards observed on the surface, for example, DOI: 10.2478/amsc-2014-0067 or DOI 10.24425/ams.2020.132712.

Best regards

Author Response

Dear reviewer, 

We would like to thank you for all your recommendations, which help to improve the quality of the manuscript. All corrections are marked in the body of manuscript by red color. I would like to point out, that because there were a lot of references added, all Reference part was rewritten and renumbered.

Specific comments to your recommendations are listed below:

I like the article. The authors did a good research-work and properly presented the results of their studies. The use of both soil tests and airborne observations were appropriate.

Thank you very much! We appreciate this.

I have only a few comments to the article:
[line 35]: In my opinion, the sentence is too general. Please explain which ones are potential and which are serious. 
Thank you for your comment. In Slovakia, all environmental loads are divided into three categories – environmental loads (the leakage of risk elements is scientifically confirmed), potential environmental loads (there is a possibility, they could be dangerous for the environment, but no scientific research was made there – or they become dangerous when disrupted) and recultivated environmental loads. A better explanation was added to the manuscript.
[line 35]: [1-3] ... I propose to add examples of works from other post-mining environmental hazards, e.g. doi.org/10.1016/j.ijrmms.2020.104377. 
Thank you for your recommendation. The reference was added to the manuscript.
[line 51]: I propose to add a reference to the described examples of mining hazards observed on the surface, for example, DOI: 10.2478/amsc-2014-0067 or DOI 10.24425/ams.2020.132712.
Thank you for your recommendationation. The references were added to the manuscript.

Reviewer 4 Report

Review of  “Evaluation of Soil and Ambient Air Pollution around Unreclaimed Mining Bodies in Post-Mining Area”

Major comments:

Overall Demová et al have conducted an interesting study on trace metal accumulation in soils, lichen, and moss around a former mining area. Their results show that dust may be an important transport mechanism to surrounding areas, which are not enriched in several trace metals. To determine if concentrations are enriched, they utilized several ratios to compare accumulation vs native concentrations. I think it is a great study and should be published with some revisions.

My only content concern is that not all trace metals should be treated the same as the authors have done here. Manganese is an essential element and can accumulate to high concentrations in soils and plants without being toxic (see Herndon et al., 2011 Environmental Science and Technology or Jordan et al., 2019 Environmental Geochemistry and Health). Thus, stating it is heavily enriched could be due to plants and could have no implications for ecosystem or human health. A similar argument can be made for Iron being a widely abundant element in soils as they develop.

Specific comments

Abstract:

Line 16: Rephrase first sentence to begin with describing the mine and why sampling is needed. Is it to explore current/active contamination from the mine or historical/previously existing contamination?

Line 26: Briefly describe how are they changing. Composition? Abundance?

Introduction:

Line 34: Change “was” to “has been”.

Line 35: Change “another is a serious environmental problem” to “others are serious environmental problems”.

Line 38: Change ‘heavy metals’ to ‘toxic metals’ as it is an antiquated term that is non-specific.

Line 47: Change “A lot of authors” to “Many previous studies”.

Lines 50 – 54: Move these sentences to the first paragraph where mining is first discussed.

Line 56 – 59: I recommend adding a sentence describing why soil microbial communities are of importance and adding another sentence describing why bioindicators are effective/useful measures of soil quality.

Materials and Methods:

Line 60: Add Köppen classification for study area.

Line 101: Describe digestion process for soil, moss and lichen samples.

Line 107: Change “Exchange soil reaction was measured by pH” to “Exchangeable soil pH was measured by pH”.

Line 108: Correct “CaCl2” to “CaCl2”.

Line 144: What was the reference element?

Figure 1: Add scale bars to the maps. How many kilometers/meters are things apart?!

Line 160: Change “death” to “dead” parts.

Results and Discussions

Line 202: Change “the area was the most” to “the area that was the most”.

Line 260: I would find additional comparisons of these enzyme results with other studies will help provide context with the findings. Can other things affect enzymes other than the toxic metals. I don’t believe organic matter content was discussed at all.

Figure 4 and 6: Please describe what are the error bars are in the figure caption. For example, are they 1 standard deviation or 1 standard error?

Line 340: Remove “air” as it is redundant.

Author Response

Dear reviewer, 

We would like to thank you for all your recommendations, which help to improve the quality of the manuscript. All corrections are marked in the body of manuscript by red color. I would like to point out, that because there were a lot of references added, all Reference part was rewritten and renumbered.

Specific comments to your recommendations are listed below:

Major comments:

Overall Demová et al have conducted an interesting study on trace metal accumulation in soils, lichen, and moss around a former mining area. Their results show that dust may be an important transport mechanism to surrounding areas, which are not enriched in several trace metals. To determine if concentrations are enriched, they utilized several ratios to compare accumulation vs native concentrations. I think it is a great study and should be published with some revisions.

My only content concern is that not all trace metals should be treated the same as the authors have done here. Manganese is an essential element and can accumulate to high concentrations in soils and plants without being toxic (see Herndon et al., 2011 Environmental Science and Technology or Jordan et al., 2019 Environmental Geochemistry and Health). Thus, stating it is heavily enriched could be due to plants and could have no implications for ecosystem or human health. A similar argument can be made for Iron being a widely abundant element in soils as they develop.

Thank you for your comment and we agree that these elements are not primarily considered toxic. But in the evaluated area, the values of manganese and iron are extremely high. Such high concentrations can have a negative/toxic effect on plants, animals, humans and also on other components of the environment.

Specific comments

Abstract:

Line 16: Rephrase first sentence to begin with describing the mine and why sampling is needed. Is it to explore current/active contamination from the mine or historical/previously existing contamination?
Thank you for your comment. The sentence was changed.

Line 26: Briefly describe how are they changing. Composition? Abundance?
Thank you for your comment. Usually, soil pollution leads to the decrease of activity of soil enzyme, but it is individualized depending on the particular enzyme. The sentence was completed.

Introduction:

Line 34: Change “was” to “has been”. Thank you. The sentence was corrected. Additionally, the sentence has been transformed on the recommendation of another reviewer

Line 35: Change “another is a serious environmental problem” to “others are serious environmental problems”. The sentence was changed.

Line 38: Change ‘heavy metals’ to ‘toxic metals’ as it is an antiquated term that is non-specific.
Thank you. The sentence was changed.

Line 47: Change “A lot of authors” to “Many previous studies”.
The sentence was changed.

Lines 50 – 54: Move these sentences to the first paragraph where mining is first discussed. The paragraph was moved according to the recommendation.

Line 56 – 59: I recommend adding a sentence describing why soil microbial communities are of importance and adding another sentence describing why bioindicators are effective/useful measures of soil quality. Thank you. Information about soil microbial communities importance and about bioindicators effectiveness was added to the paragraph.

Materials and Methods:

Line 60: Add Köppen classification for study area.
Thank you. The Köppen classification was added to the Methodology part.

Line 101: Describe the digestion process for soil, moss and lichen samples.
As we mentioned in the Methodology part, assessment of the hazardous elements in the soil, moss and lichen samples was several times described in detail in our earlier studies [17, 23, 28, 29]. This section is supported by references, where you can find the digestion process described in detail.

Line 107: Change “Exchange soil reaction was measured by pH” to “Exchangeable soil pH was measured by pH”.
The sentence was corrected according to your recommendation.

Line 108: Correct “CaCl2” to “CaCl2”.
The mistake was corrected.

Line 144: What was the reference element?
Thank you. Information about the reference element was added to the sentence.

Figure 1: Add scale bars to the maps. How many kilometers/meters are things apart?!
Thank you for your recommendation. The scales were added to the maps.

Line 160: Change “death” to “dead” parts.
The sentence was corrected.

Results and Discussions

Line 202: Change “the area was the most” to “the area that was the most”.
Thank you for your comment. The sentence was corrected.

Line 260: I would find additional comparisons of these enzyme results with other studies will help provide context with the findings. Can other things affect enzymes other than the toxic metals. I don’t believe organic matter content was discussed at all.
Thank you for your recommendation. Discussion about the influence of soil parameters on the activity of soil enzymes was extended.

Figure 4 and 6: Please describe what are the error bars are in the figure caption. For example, are they 1 standard deviation or 1 standard error?
Boxplots in Figure 4 and Figure 6 are explained in the legend. The upper part of the box shows the average value. As you mentioned, there was a possibility to chose a boxplot, where the box express standard deviation, and lines (min-max in our case) express quartiles. But it is not our case (we didn't choose this type of chart)

Line 340: Remove “air” as it is redundant.
Air” was removed from the sentence.

Best regards, 

Round 2

Reviewer 1 Report

The authors attempted to make the amendment of most of the comments in order to improve the manuscript.

The manuscript is to be accepted subject to the following - 

  1. Some more editing and proofreading are required to improve the presentation and clarity.
  2. Re Lines 364-367 on the conclusion, the "dangerous" part still deviates from the results of the previous health risk assessment. It has to be checked again and revised accordingly. 

Author Response

Dear reviewer, 

Thank you again for all your recommendations.

  1. Some more editing and proofreading are required to improve the presentation and clarity.
    The manuscript was properly checked. Several stylistic and grammatical errors were found and corrected. All changes are marked in the manuscript.

  2. Re Lines 364-367 on the conclusion, the "dangerous" part still deviates from the results of the previous health risk assessment. It has to be checked again and revised accordingly.

    We wanted to say that despite the fact that the consumption of mushrooms proved to be safe, the study area is still dangerous to human health in terms of soil pollution (because of the production of the crop, and its consumption by inhabitants) and air pollution, which was also found by our results as serious. Soil and air pollution have been proven by several authors to be risky in terms of public health. Given our results, we cannot consider the area safe for human health.

Best regards, 
